# Feed Efficiency, Tissue Growth and Energy Budget Changes during the Molting Cycle of Juvenile Mud Crab, *Scylla serrata*: Effects of Dietary Proteins, Fishmeal versus Soy Protein Concentrate

Ngoc Thi Bich Nguyen [1], Laurent Wantiez [2,*], Pierrette Lemaire [3] and Liet Chim [3]

1 RIA3, Research Institute for Aquaculture No 3, Nha Trang 57000, Vietnam
2 UMR Entropie, University of New Caledonia, Nouméa 98800, New Caledonia
3 IFREMER, Délégation de Nouvelle-Calédonie, Nouméa 98800, New Caledonia
* Correspondence: laurent.wantiez@unc.nc

**Abstract:** Three isoenergetic diets differing in their fishmeal/soy protein concentrate (SPC) ratio were assessed on the tissue growth and energy budget of juvenile crabs *Scylla serrata* in postmolt stages (PMolt) and in intermolt stages (IMolt). The average growth rates on a dry matter basis were $2.064 \pm 0.324\%$ and $0.492 \pm 0.08\%$ initial BW.day$^{-1}$ during PMolt and IMolt stages, respectively. The efficiencies of the feed conversion (FCE, %), protein retention (PRE, %) and energy retention (ERE, %) were similar for the three experimental diets. However, FCE, PRE and ERE in PMolt stages were four to five times higher than in IMolt stages. The feed intake, energy and protein required for growth in PMolt stages were obviously higher than in IMolt stages. The energy budgets (% total energy intake) were marginally affected by diet but were significantly affected by the molt stage. The maintenance energy was lower in PMolt stages ($49.84 \pm 4.9\%$) than in IMolt stages ($83.33 \pm 2.45\%$). The excess in maintenance energy in IMolt stages represents the portion set aside for the next molt: shell energy content ($4.97 \pm 0.31\%$) and energy for ecdysis ($\pm 28\%$). Conversely, recovery energy was significantly higher in PMolt stages ($34.39 \pm 0.99\%$) than in IMolt stages ($8.33 \pm 1.7\%$). In conclusion, SPC sustained good tissue growth and good feed utilization and can be used as a main source of dietary protein for crab juveniles in captivity.

**Keywords:** fish meal; soy protein concentrate; feed efficiency; tissue growth; protein requirement; energy budget; juvenile crab; *Scylla serrata*

## 1. Introduction

The mud crab has been domesticated and farmed for many years in Asia, particularly in Vietnam, where production is increasing quickly (www.shrimpnews.com). The mud crab of the genus *Scylla* are being extensively cultured in the Asia-Pacific region due to their high market price [1,2]. However, the substantial crab farming operations that exist are still mainly based on trash fish, which contaminate rearing water and lead to highly cultured water pollution.

In New Caledonia, there is a political wish to diversify aquaculture, which has been based on blue shrimp (*Litopenaeus stylirostris*) farming. Among the different options available, the mud crab, the local species *Scylla serrata*, is regarded as having potential for aquaculture through its farming aptitudes and its high economical value. However, the possible development of crab farming in New Caledonia necessarily involves the development of an artificial feed that can be produced from selected and controlled raw materials available on the international market. Therefore, it is essential to assess the nutritional requirements of *S. serrata*. A few studies have been conducted on this subject:

Sheen and Wu [3] showed better lipid utilization by the crab than shrimp by evaluating several amounts of a mixture of cod liver oil and corn oil on the growth of juvenile crabs.

The same authors have shown the importance of a dietary source of cholesterol and polyunsaturated fatty acids (22:6 n-3, 20:4 n-6, 18:3 n-3) for the healthy growth of juvenile crabs [4].

Furthermore, a series of studies have been conducted to measure the digestibility of several potential ingredients in formulated feed for crabs [5,6]. The results showed that crabs were able to digest many different ingredients, in particular, fiber and protein from plants.

Finally, the optimal protein inclusion level in the diet has been the subject of few studies [7–9]. These studies concluded that the optimal protein concentration (a mixture of fishmeal and soybean meal) in the feed for the best growth rate is between 30% and 47%.

Most of the previous studies have considered the growth rate as gains in fresh weight. However, fresh weight changes follow a basic pattern through the molt cycle, i.e., large and abrupt increases associated with rapid water uptake at ecdysis; further moderate gains associated with carapace mineralization and tissue growth during postmolt; and the relative stabilization of fresh weight during intermolt until the onset of the successive ecdysis [10]. An increase in tissue mass, on the other hand, is a continuing process occurring through the molt cycle [9,11]. Tissue growth can be measured as the increase in dry mass of the animal; this measure is more relevant than live mass, which is subject to the extremely variable water content of the molting cycle in crustaceans [9,12].

Our paper presents the experimental results that allow us to access the effects of two protein sources (fishmeal vs. soy protein concentrate) on the voluntary feed intake, tissue growth, fecal production, nitrogenous excretion and energy budget of juvenile crabs during postmolt and intermolt periods. Thus, our results provide a better understanding of the effect of the protein sources on the tissue growth and dietary energy utilization of juvenile crabs during a complete molting cycle. This information is fundamental to adjusting the feed formula and ration level for the balanced nutrition of the captive aquaculture juveniles *S. serrata*. Finally, based on the data collected during this study, our previous study [9] and the literature [10], we propose a schematic growth model for juvenile *S. serrata* over two molting cycles as a function of the quantity of feed ingested.

## 2. Materials and Methods

### 2.1. Crabs and Holding Facilities

Juvenile crabs, *Scylla serrata*, were collected in the shallow intertidal zones associated with mangroves surrounding the experimental station (21°51′50″ S, 166°3′0″ E) in Boulouparis district, New Caledonia. Eighty crab juveniles were caught using a hand net and transferred to the laboratory and acclimated into 6 circular composite tanks (capacity 500 L) for two weeks prior to the beginning of the experiment. Next, sixty of these crabs were identified for their molt stages based on the criteria and methodology described by [13,14]. These crabs were randomly, individually assigned to sixty experimental rectangular polyethylene tanks (30 cm × 20 cm × 30 cm) covered with black lids.

Two experimental crab groups, with 30 individuals per each group, were considered: postmolt (PMolt) and intermolt (IMolt). At the beginning of the experiment, the PMolt group was constituted of crabs that were about to molt (from D1 to D3 molt stages), and the IMolt group included crabs that were in early intermolt stages (C3 to C4).

During the trial, the PMolt crabs molted within the first two weeks and reached C2 or C3 stage; conversely, the IMolt crabs did not molt, and they attained D2 or D3 stage at the end of the experiment.

After being dried in soft paper, each crab was weighed to the nearest 0.01 g and its carapace width was measured to the nearest 0.01 mm.

Each experimental tank was continuously supplied with seawater running from a plastic reservoir (2000 L capacity) at a rate of 0.19 L·min$^{-1}$. The water pumped from the lagoon was filtered through a 25 μm net bag into the reservoir. The temperature was automatically controlled using a heater. The temperature in the experimental tanks was measured continuously (every 3 h) using an automatic recording probe. During the

experimental period, temperature, salinity, pH and dissolved oxygen in the water were maintained at 28.5 ± 1.5 °C, 34.0 ± 1.5‰, 7.87 ± 0.09 and 4.46 ± 0.16 mg·L$^{-1}$, respectively.

## 2.2. Diet Preparation and Composition

The crabs were fed on experimental diets produced in the laboratory; the ingredients were ground up in a grinder (Retsch®, Haan, Germany) with a 1 mm screen. Next, the meal obtained was mixed with oil and water (30%) in a horizontal mixer (Mainca®, Barcelona, Spain) until the consistency was suitable for pelleting. The mixture was then extruded through a 3 mm die in a meat grinder. Pellets were then steamed for 15 min and stored at −20 °C before use. The ingredient composition and proximate nutrient content of experimental diets are shown in Table 1.

**Table 1.** Proximal composition of ingredients and three experimental diets.

| | Diets | | |
|---|---|---|---|
| | **F100** | **F50/S50** | **S100** |
| Composition (%) | | | |
| Fish meal | 53.28 | 26.64 | |
| Soy protein concentrate | | 28.70 | 57.39 |
| Crab meal | 19.59 | 19.59 | 19.59 |
| Fish oil | | 1.96 | 3.92 |
| Wheat | 16.65 | 13.42 | 10.77 |
| Limestone | 6.46 | 5.68 | 4.31 |
| Dicalcium phosphate | 1.96 | 1.96 | 1.96 |
| Binder [1] | 0.59 | 0.59 | 0.59 |
| Vitamin premix [2] | 0.29 | 0.29 | 0.29 |
| Vitamin C [3] | 0.88 | 0.88 | 0.88 |
| Trace elements | 0.29 | 0.29 | 0.29 |
| Analysis (dry matter basis) | | | |
| Crude protein—P (%) | 49.60 | 48.81 | 48.39 |
| Lipid (%) | 4.98 | 4.69 | 4.19 |
| Ash (%) | 23.82 | 21.73 | 16.22 |
| Gross energy—E (MJ·kg$^{-1}$) | 15.08 | 15.41 | 15.34 |
| P/E (g·Mj$^{-1}$) | 32.89 | 31.68 | 31.54 |

| | Proximal Analysis (Dry Matter Basis) | | | |
|---|---|---|---|---|
| | **Protein (%)** | **Lipid (%)** | **Ash (%)** | **Energy (MJ·kg$^{-1}$)** |
| Composition (%) | | | | |
| Fish meal [4] | 73.73 | 11.18 | 12.40 | 22.47 |
| Soy protein concentrate [5] | 64.02 | 2.47 | 7.00 | 23.53 |
| Crab meal | 42.23 | 1.90 | 46.47 | 11.47 |
| Fish oil | | | | 39.27 |
| Wheat | 12.92 | 2.13 | 2.02 | 15.73 |

[1] Pegabind (R) synthetic resin from Bentoli Cie. [2] Vitamin premix for shrimp from BEC feed solutions PTY. LTD ingredients: vitamin AD3 1000/200, vitamin B1 thiamine 98%, mononitrate, vitamin B2 riboflavin 80%, vitamin B3 niacin 99%, vitamin B5 D-Calpan 98%, vitamin B6 pyridoxine 98%, vitamin B9 folic acid 97%, vitamin D3 500, vitamin E 50 ADD, vitamin K3 43.7%. [3] Vitamin C: Rovimix Stay-C 35 from DSM Cie. [4] Peruvian fish meal and fish oil (SGS del Peru S.A.C.). [5] SOYPRO processed by Bio Processing.

## 2.3. Experimental Design

### 2.3.1. Growth Trial

Table 2A shows the experimental design of this study. Sixty (60) tanks had individual crabs that were assigned to two groups of 30 individuals: PMolt and IMolt. As each crab was separately raised, the experimental unit in this study was the individual. Each crab group was fed on three different experimental diets distributed ad libitum: F100; F50S50; and S100. Thus, 10 tanks of each crab group were randomly assigned to each diet (n = 10).

**Table 2.** Experimental design of trials: growth (A), digestibility (B), fecal production (C) and nitrogenous excretion (D).

| Trials | Experimental Diets | | | | | | | | | | | | | | |
|---|---|---|---|---|---|---|---|---|---|---|---|---|---|---|---|
| | **F100** | | | | | **F50S50** | | | | | **S100** | | | | |
| **(A) Growth** | | | | | | | | | | | | | | | |
| Ration size (% iBW d$^{-1}$) | AL | | | | | AL | | | | | AL | | | | |
| Number of PMolt crabs | 10 | | | | | 10 | | | | | 10 | | | | |
| Number of IMolt crabs | 10 | | | | | 10 | | | | | 10 | | | | |
| Feeding frequency | 1/24 h | | | | | 1/24 h | | | | | 1/24 h | | | | |
| **(B) Digestibility** | | | | | | | | | | | | | | | |
| Ration size (% iBW d$^{-1}$) | Ad libitum | | | | | Ad libitum | | | | | Ad libitum | | | | |
| Replicates (n) | 5 | | | | | 5 | | | | | 5 | | | | |
| Feeding frequency | 2/24 h | | | | | 2/24 h | | | | | 2/24 h | | | | |
| **(C) Fecal production** | | | | | | | | | | | | | | | |
| Ration size (% iBW d$^{-1}$) | 0.5 | 1.0 | 1.5 | 2 | AL | 0.5 | 1.0 | 1.5 | 2 | AL | 0.5 | 1.0 | 1.5 | 2 | AL |
| Number of PMolt crabs | | | 10 | | | | | 10 | | | | | 10 | | |
| Number of IMolt crabs | | | 10 | | | | | 10 | | | | | 10 | | |
| Feeding frequency | | | 1/24 h | | | | | 1/24 h | | | | | 1/24 h | | |
| **(D) Nitrogenous excretion** | | | | | | | | | | | | | | | |
| Ration size (%iBW d$^{-1}$) | 0.5 | 1.0 | 1.5 | 2 | uf | 0.5 | 1.0 | 1.5 | 2 | uf | 0.5 | 1.0 | 1.5 | 2 | uf |
| Replicates (n) | 3 | 3 | 3 | 3 | 3 | 3 | 3 | 3 | 3 | 3 | 3 | 3 | 3 | 3 | 3 |

AL: Ad libitum; uf: unfed. PMolt crabs included crab juveniles in premolt stage (from D1 to D3) at the beginning of the experiment. IMolt crabs included crab juveniles in early intermolt stage (from C3 to C4) at the beginning of the experiment.

At the beginning of the experiment, the crabs were fasted for 48 h and then weighed and measured. The initial average body weight and carapace size of crabs were $22.10 \pm 8.46$ g and $5.06 \pm 0.70$ cm, respectively. The trial was conducted over 32 days. At the end of the experiment all crabs were left for 48 h starvation and then weighed, measured and dried at 80 °C for 48 h, and then kept at −20 °C before biochemical and energy content analysis.

The daily pre-weighed feed rations for each tank were delivered once a day at 6:00–7:00 am. Three hours after each meal, the unconsumed feed was siphoned off, dried for 24 h at 80 °C and then weighed. The leaching rate (dry matter loss of the pellet in the water) was determined by measuring the remaining dry matter of the diet after 3 h immersion. Then, the amount of ingested feed (dry matter basis) was obtained following this equation:

$$FI = dF − L − uF$$

where FI = ingested Feed; dF = distributed Feed; L = leaching after 3 h; and uF = unconsumed Feed.

Finally, the specific daily feed intake (ingested diet, % iBW d$^{-1}$) of each crab was determined.

During the experimental period only crabs from PMolt group molted; the date of ecdysis, carapace width and body weight were recorded for each crab after molting. The exuviae was collected and weighed before and after drying at 80 °C for 24 h, and then kept at −20 °C for energy analysis.

### 2.3.2. Digestibility Trial

The experimental design of this trial is presented in Table 2B. All diets contained 1% chromic oxide ($Cr_2O_3$) as an inert indicator to allow the calculation of digestibility coefficients (ADC) for dry matter (ADMD), crude protein (ACPD) and gross energy (AGED).

For each experimental diet, five crabs were fed ad libitum twice daily until approximately 1.0 g of fecal material (dry weight) was collected (Table 2B). Crab feces at the bottom of the tank were collected individually by Pasteur pipette and rinsed gently in distilled water. Fecal matter from five crabs in each experiment diet was pooled. All samples were kept at −20 °C before freeze drying for analysis.

The chromium content of diets and fecal material used to calculate apparent digestibility values were determined using the method described by [15]. Apparent digestibility of dry matter (ADMD), crude protein (ACPD) and gross energy (AGED) were calculated using the following equation:

$$ADMD = 100 - 100 \times (\% \, Cr_2O_3 \text{ in feed}/\% \, Cr_2O_3 \text{ in feces})$$

The digestibility of crude protein (ACPD) or gross energy (AGED) was determined using the formula ACPD or AGED $= 100 - 100 \times (\% \, Cr_2O_3 \text{ in feed}/\% \, Cr_2O_3 \text{ in feces}) \times$ (% protein or MJ kg$^{-1}$ energy in feces/% protein or MJ kg$^{-1}$ energy in feed).

### 2.3.3. Fecal Production Trial

The experimental design of this trial is presented in Table 2C. Sixty tanks with individual crabs were assigned to PMolt and IMolt groups of 30 individuals each. Each group was randomly divided into 3 subgroups of 10 individuals, each subgroup receiving the three different experimental diets (F100; F50S50; and S100). Diets were distributed in 5 ration sizes (% of initial fresh body weight = % iBW): 0.5; 1; 1.5; 2; and ad libitum. Each ration size was applied to 2 tanks for each diet treatment.

Fecal production was collected for each experimental unit three times a day across 10 days (Table 2C). Then, the feces were pooled for each diet treatment and whatever the ration size. At the end of the trial, the collected fecal matter was freeze-dried, weighed and stored at $-20\,°C$ before biochemical and energy analysis.

Fifteen crabs were individually fed at five different ration sizes for each experimental diet (three replicates for one treatment): unfed; 0.5; 1; 1.5; and 2% (Table 2D). Immediately after the meal, each crab was transferred into another individual aquarium with fixed water volume (30 L). Ammonia-N excretion [16] in each aquarium was measured twice according to the method described in [17]: before and 24 h after transferring the crab. Then, the ammonia-N value was converted into energy by multiplying by 24.83 kJ·g$^{-1}$ [18]. The potential loss of nitrogenous compounds through bacterial action or diffusion was quantified by setting a controlled aquarium without crabs. The values of ammonia-N concentration of this control treatment were used to adjust determined nitrogenous excretion.

### 2.4. Biochemical Analysis

The proximate compositions of diets, feces, initial and final crabs were estimated according to the method of [19] in an in vivo laboratory, Vietnam. Moisture contents of crabs and feed were determined after oven-drying to a constant weight at 105 °C. Each sample was combusted at high temperature in pure oxygen; then, the nitrogen content was measured by thermal conductivity detection and converted to equivalent protein by an appropriate numerical factor: crude protein % = % N $\times$ 6.25. Crude lipid content was measured by solvent extraction with petroleum ether. Ash content was determined by a muffle furnace at 550 °C for 8–10 h. Energy contents of diets, fecal material, exuviae and crabs were determined by calorimetric bomb (Parr® 6200, Jasper, IN, USA, calibrated by benzoic acid) at Ifremer laboratory, New Caledonia.

### 2.5. Definition, Calculation and Statistics

The energy partitioning of the budget suggested by the National Research Council [20] was adopted in this study with minor modification, in order to comply with mud crab growth through molting. The energy budget of growing crabs is expressed in our study as:

$$IE = FE + UE + SE + RE + HEm \tag{1}$$

where:

IE is the energy intake that was obtained for each crab by calculating the difference between the feed distributed and the feed leached and uneaten.

FE is energy lost from the animal through feces for each crab; FE was calculated from the model linking the feed intake (energy intake) and fecal production.

UE is energy lost from the animal through the total ammonia excretion, which was calculated for each crab from the model linking the feed intake (energy intake) and nitrogenous excretion.

SE is surface energy losses, which is exuvia lost at ecdysis for the PMolt crab group.

RE is the recovered energy, which is channeled into growth. RE was estimated considering the initial crab energy content calculated from the relationship defined between dry and fresh weight ($Y = 0.3229 X + 0.4246$; $R^2 = 0.96$, $n = 30$). It was calculated for each crab through the dry weight gain converted into energy.

HEm is the maintenance energy calculated for each crab from the Equation (1):

$$HEm = IE - (FE + UE + SE + RE) \tag{2}$$

Specific tissue growth rate (SGRd, % dry iBW.day$^{-1}$) = $100 \times$ (dry fBW $-$ dry iBW) $\times$ dry iBW$^{-1}$.day$^{-1}$; where fBW and iBW are final and initial body weight, respectively.

Initial dry body weight (dry iBW) was estimated through equation: $Y = 0.3229 X + 0.4246$ ($R^2 = 0.96$, $n = 30$) where Y is dry body weight and X is fresh body weight. This equation was obtained from our extra sample assessment for crabs in intermolt (IMolt group). In our experiment, we showed in our extra sample assessment that the dry body weight of individuals from PMolt group did not change significantly before and after ecdysis. Therefore, in our study, for PMolt crabs, we used the body weight after ecdysis as the initial body weight (iBW). Consequently, for PMolt crab group, the growth period study lasted between ecdysis and the end of the experiment with total duration of 18 to 22 days, while for IMolt crabs the growth period was the same as the experimental duration at 32 days.

The efficiency in feed conversion (FCE), protein retention (PRE) and energy retention (ERE) is the ratio between weight gain in dry matter, protein and energy and the amount of feed (dry matter basic), protein or energy ingested, respectively [20–24].

Nitrogenous excretion (UE), fecal production (FE), recovered energy (RE) and maintenance energy (HEm) were analyzed using ANCOVA analysis with feed intakes as the covariate. For these parameters, the adjusted individual data and mean were calculated for the same amount of ingested feed.

Optimum growth is defined as the best increase in dry body weight for the least feed intake [25]. Therefore, the feed intake corresponding to the highest values of FCE was determined as the optimum intake, and then converted into protein and energy required for optimal tissue growth of the individual crab.

Statistical analyses were performed on SPSS 16.0 (IMB, New York, NY, USA) for Microsoft Windows. There were significant differences among the various treatments when $p < 0.05$. Shapiro–Wilk and Levene's test were used for checking normality and homogeneity of variances, respectively. Percentage data were transformed into arcsine to get normality. Two-way ANOVA fixed model was used to test the effects of diet and crab group (PMolt or IMolt) on body biochemical composition, energy content and each energy budget component (FE, UE, Hem and RE). Analyses of covariance (ANCOVA) were used to investigate the effect of different experimental diets on nitrogenous excretion, fecal production and specific growth rate across the amounts of feed ingested. Linear function between nitrogenous excretion and feed intake was applied for ANCOVA analysis, while the best relationship was exponential. Data were Log-transformed to get a linear relation between growth rate and feed intake for IMolt crabs. Digestibility coefficients of crabs fed on different experimental diets were compared by one-way ANOVA.

## 3. Results

### 3.1. Body Chemical Composition and Energy Content

The contents of protein, lipid, ash and energy in the body of PMolt and IMolt crab juveniles for the three different experimental diets are presented in Table 3. Whatever the

crab group, PMolt or IMolt, no significant effect of the diet was shown on the biochemical composition and energy content of the animal ($p$ = 0.29; 0.31; 0.74; and 0.13 for protein, lipid, ash and energy, respectively).

**Table 3.** Crab body proximal composition (%) and energy content (kJ·g$^{-1}$ of dry body weight) on dry weight basis in PMolt and IMolt crab groups fed ad libitum.

| Group | Diet | Protein | Lipid | Ash | Energy |
|---|---|---|---|---|---|
| PMolt | F100 (n = 3) | 38.7 ± 1.1 [a] | 1.5 ± 0.1 [a] | 46.2 ± 2.0 [a] | 10.4 ± 1.1 [a] |
| | F50S50 (n = 3) | 37.6 ± 2.5 [a] | 1.3 ± 0.8 [a] | 46.3 ± 4.3 [a] | 9.9 ± 1.1 [a] |
| | S100 (n = 3) | 41.9 ± 4.2 [a] | 1.9 ± 0.4 [a] | 42.9 ± 3.4 [a] | 11.2 ± 0.7 [a] |
| IMolt | F100 (n = 3) | 39.9 ± 4.7 [a] | 2.8 ± 0.8 [b] | 42.2 ± 3.3 [a] | 10.1 ± 0.1 [a] |
| | F50S50 (n = 3) | 39.7 ± 1.3 [a] | 2.4 ± 0.1 [b] | 42.5 ± 0.9 [a] | 10.0 ± 0.3 [a] |
| | S100 (n = 2) | 42.3 ± 1.5 [a] | 3.1 ± 0.4 [b] | 40.0 ± 0.3 [a] | 11.5 ± 1.1 [a] |

Values are mean ± SD (n = replicates per treatment), within the same column, means with different letters are significantly different ($p$ < 0.05).

During the study, the body lipid content was significantly higher in IMolt than in PMolt crabs ($p$ = 0.003) for the three diets tested.

### 3.2. Diet Digestibility

Apparent digestibility for dry matter (ADMD), crude protein (ACPD) and gross energy (AGED) across the three experimental diets are given in Table 4. No significant difference in ADMD, ACPD and AGED was found between the diets ($p$ = 0.43; 0.59; and 0.19, respectively). The averages of ADMD, ACPD and AGED were high for each diet: 90.34 ± 1.79%; 97.72 ± 0.44%; and 90.42 ± 1.82, respectively.

**Table 4.** Apparent digestibility (%) for dry matter (ADMD), crude protein (ACPD) and gross energy (AGED) of three experimental diets.

| Diets | Digestibility | | |
|---|---|---|---|
| | ADMD | ACPD | AGED |
| F100 | 90.56 ± 2.38 [a] | 97.41 ± 0.65 [a] | 89.06 ± 2.76 [a] |
| F50S50 | 91.03 ± 1.85 [a] | 97.86 ± 0.44 [a] | 92.27 ± 1.59 [a] |
| S100 | 89.44 ± 1.15 [a] | 97.88 ± 0.23 [a] | 89.92 ± 1.1 [a] |

Values are mean ± SD (n = 3 replicates per treatment), within the same column, means with the same letters are not significantly different ($p$ > 0.05).

### 3.3. Fecal Production and Nitrogenous Excretion

The fecal production of juvenile crabs, whatever experimental crab group was considered, PMolt or IMolt, increased significantly with feed intake ($p$ < 0.01) for all three experimental diets. The relationship between fecal production and feed intake was described as a simple linear function (Figure 1). The diet composition did not affect fecal production ($p$ = 0.46).

The nitrogenous excretion of juvenile crabs increased with feed intake. Exponential functions were used to describe this relationship for all three experimental diets (Figure 2). The feed type and feed intake significantly affected nitrogenous excretion ($p$ < 0.01). The latter was significantly lower for F100, intermediate for S100 and higher for F50S50 (pairwise comparisons, $p$ < 0.01).

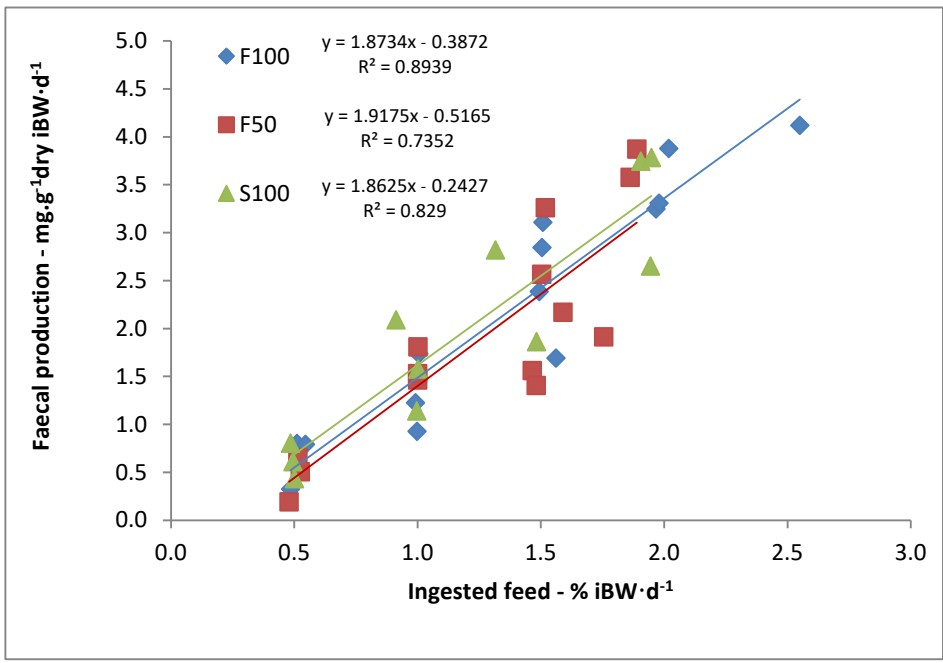

**Figure 1.** Relationships between fecal production of juvenile crabs and ingested feed for the three experimental diets. Amount of feed intake is percentage of initial fresh body weight (iBW) per day.

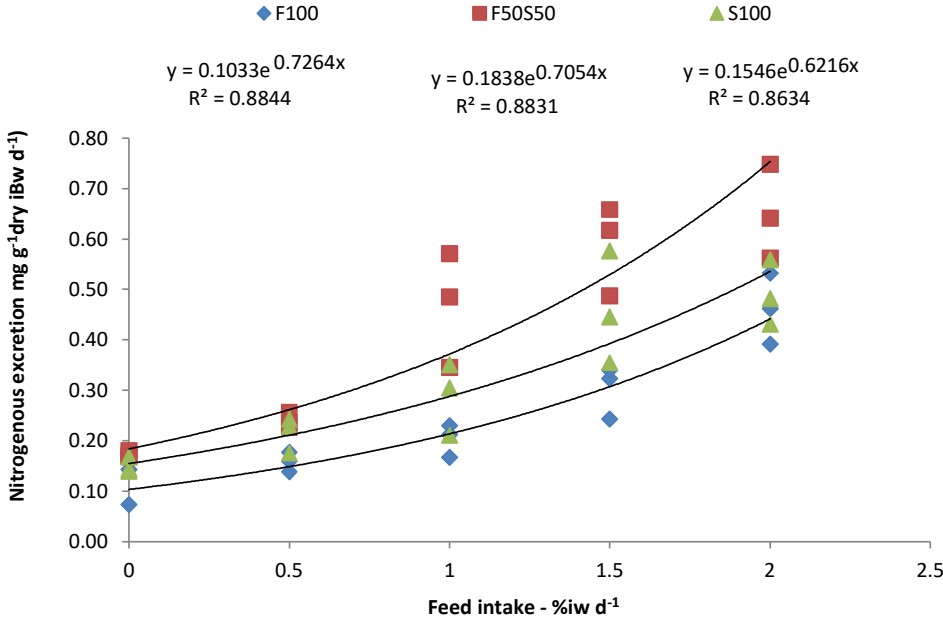

**Figure 2.** Relationships between fecal production of juvenile crabs and ingested feed for the three experimental diets. Amount of feed intake was percentage of initial fresh body weight (iBW) per day.

*3.4. Growth and Feed Utilization*

Crabs from the PMolt group molted within the first two weeks of the experiment and no crabs molted in the IMolt group; at the end of the experiment, crabs were recorded in C2 to C3 and D2 to D3 of molt stages for PMolt and IMolt groups, respectively.

The specific growth rate, in dry weight or tissue growth (SGRd), of juvenile crabs in PMolt and IMolt groups increased with feed intake ($p < 0.01$), whatever the experimental diets. Linear and logarithmic functions best fit the "tissue growth–feed intake" relationship of PMolt and IMolt crabs, respectively (Figures 3 and 4).

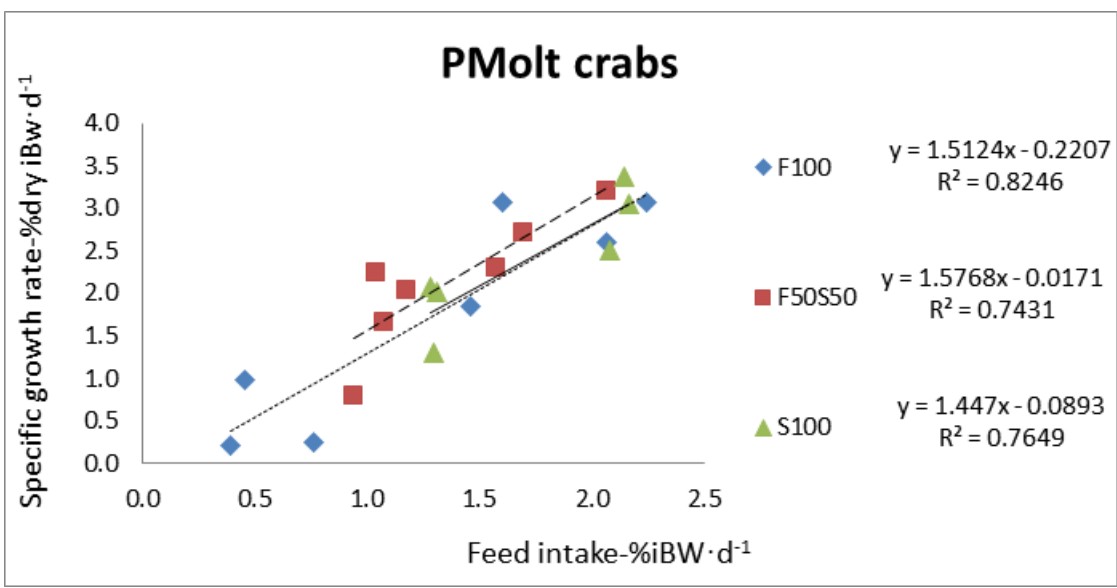

**Figure 3.** Relationship between specific growth rate (SGRd) and feed intake of juvenile crabs for the 3 experimental diets for PMolt crabs. Amount of feed intake is percentage of initial fresh body weight (iBW) per day.

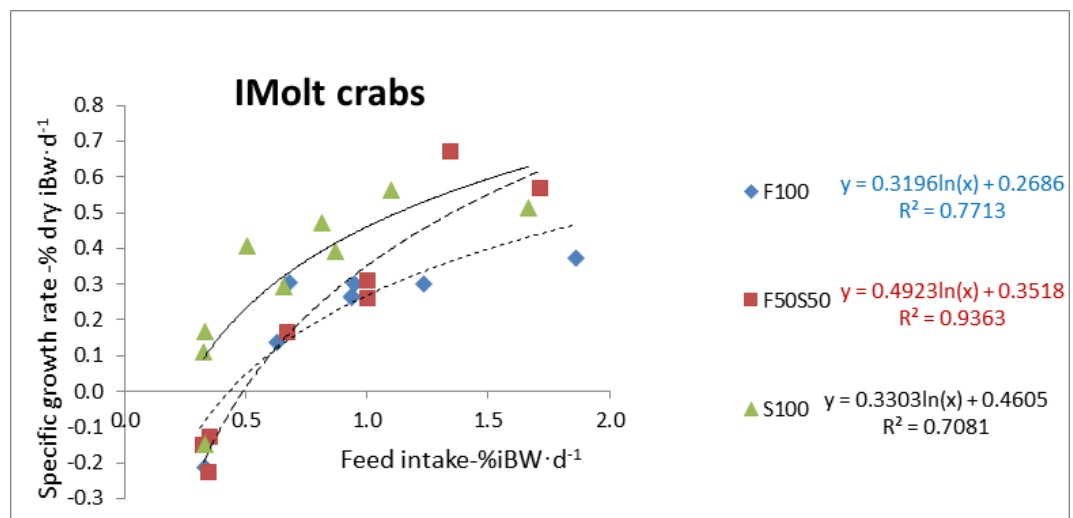

**Figure 4.** Relationship between specific growth rate (SGRd) and feed intake of juvenile crabs for the 3 experimental diets at IMolt crabs. Amount of feed intake is percentage of initial fresh body weight (iBW) per day.

ANCOVA analysis with the feed intake as a covariate allowed us to compare the adjusted means of tissue growth (SGRd) (Figure 5). For the PMolt crab group, the tissue growth was not significantly affected by the diet type ($p = 0.43$). For IMolt crabs, it was significantly higher on diet S100 compared to F100 ($p = 0.02$) and F50S50 ($p = 0.04$), but no difference was observed between diet F100 and diet F50S50 ($p = 0.98$).

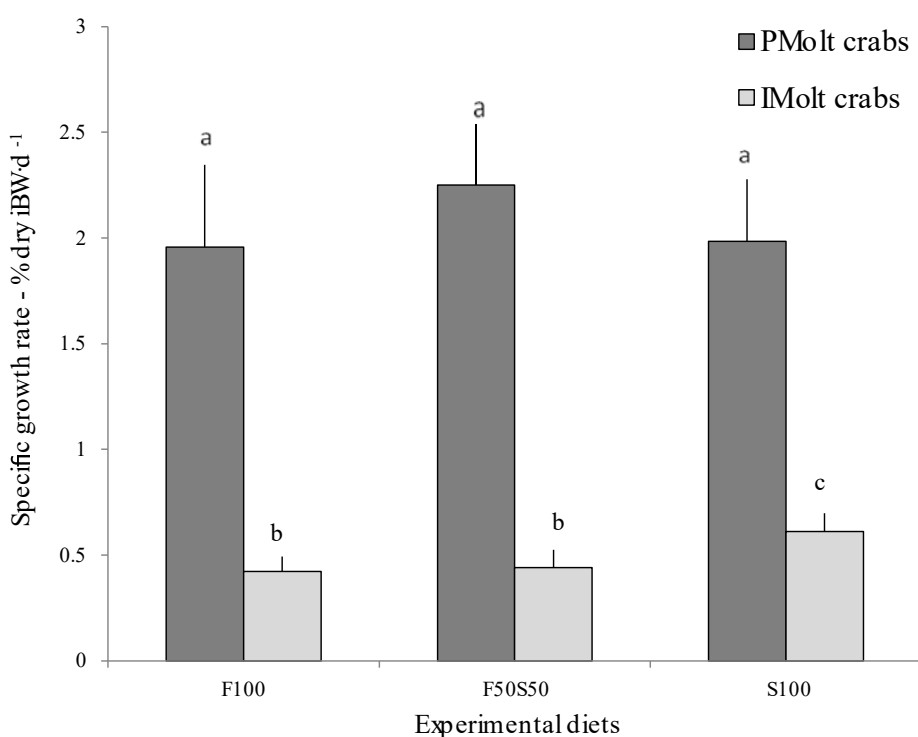

**Figure 5.** Adjusted mean specific growth rates ($\pm$ SD) for average feed intakes of PMolt and IMolt crabs. Means with different superscript letters differ significantly ($p < 0.05$).

Data presented in Figure 5 show that, during a molt cycle, 80% of the tissue growth occurred during postmolt (PMolt crabs) and only 20% during intermolt (IMolt crabs).

The feed intake values (in dry matter, energy and protein) for the optimal tissue growth (the best growth rate for the minimum feed intake) of PMolt and IMolt crabs were calculated from the polynomial functions linking feed conversion efficiency (FCE) and feed intakes:

(i) For the PMolt crabs, as FCE was not affected by the experimental diets, only one polynomial equation was defined by mixing the data (n = 16) from all three diets:

$$y = -36.296x^2 + 114.26x - 33.494 \ (R^2 = 0.65). \tag{3}$$

The calculated feed intake for optimum growth is 4.57% dry iBW·d$^{-1}$, and the derived optimum values for energy and protein intakes are 0.70 kJ. dry body g$^{-1}$·d$^{-1}$ and 0.022 g·dry body g$^{-1}$·d$^{-1}$, respectively.

(ii) For the IMolt crab group, as FCE was significantly higher in diet S100, two polynomial equations were defined: one for diet S100 treatment and one by pooling data (n = 14) for diets F100 and F50S50.

$$\text{Diet S100 (n = 7): } y = -12.362x^2 + 22.008x + 8.4093 \ (R^2 = 0.67) \tag{4}$$

$$\text{Diets [F100+F50S50] (n = 14): } y = -20.679x^2 + 54.56x - 22.728 \ (R^2 = 0.76) \tag{5}$$

The values of feed intake for optimal growth are 2.72% and 4.24% dry iBW·d$^{-1}$, respectively, for S100 and (F100 and F50S50) diets. The corresponding values of the energy and protein intakes are 0.42 kJ·dry body·g$^{-1}$·d$^{-1}$ and 0.013 g.dry g$^{-1}$·d$^{-1}$ for diet S100, and 0.68 kJ·dry body g$^{-1}$·d$^{-1}$ and 0.021 g·dry body·g$^{-1}$·d$^{-1}$ for diet (F100 and F50S50).

The estimated feed intake values for optimal growth reveal two trends: firstly, feed intake is obviously higher in the PMolt group compared to the IMolt group, and, secondly, the consumption of diet S100 was lower than that of diets F100 and F50S50 for the IMolt crab group.

The adjusted mean values, for the optimum feed intake, of the feed conversion efficiency (FCE), protein retention efficiency (PRE) and energy retention efficiency (ERE) were calculated from Equations (3)–(5) for PMolt and IMolt crab groups fed on three different diets (Table 5). The FCE, PRE and ERE values for PMolt crabs were, respectively: $49.86 \pm 7.01\%$, $39.29 \pm 5.52\%$ and $35.34 \pm 4.27\%$, and obviously 4–5 times higher than IMolt crabs, with respective values of $11.37 \pm 1.85\%$, $9.1 \pm 1.48\%$ and $7.84 \pm 1.27\%$.

**Table 5.** Adjusted mean values of efficiency in feed conversion (FCE), protein retention (PRE) and energy retention (ERE) at the optimal feed intake for crab groups, PMolt and IMolt, fed on three different diets.

| Group | Diet | Feed Efficiency | | |
|---|---|---|---|---|
| | | FCE | PRE | ERE |
| PMolt | F100 (n = 7) | $47.57 \pm 7.97$ [a] | $37.49 \pm 6.28$ [a] | $34.29 \pm 4.31$ [a] |
| | F50S50 (n = 7) | $50.39 \pm 5.29$ [a] | $39.72 \pm 4.17$ [a] | $33.03 \pm 3.41$ [a] |
| | S100 (n = 7) | $51.61 \pm 7.77$ [a] | $40.68 \pm 6.13$ [a] | $38.70 \pm 5.10$ [a] |
| IMolt | F100 (n = 5) | $7.55 \pm 0.87$ [b] | $6.04 \pm 0.70$ [b] | $5.21 \pm 0.60$ [b] |
| | F50S50 (n = 5) | $10.15 \pm 3.38$ [b] | $8.12 \pm 2.70$ [b] | $7.00 \pm 2.33$ [b] |
| | S100 (n = 6) | $16.43 \pm 1.29$ [c] | $13.14 \pm 1.03$ [c] | $11.33 \pm 0.89$ [c] |

FCE: $100 \times$ (dry final body weight $-$ dry initial body weight)/feed intake; PRE: $100 \times$ [(dry final body weight $\times$ final protein content) $-$ (dry initial body weight $\times$ initial protein content)]/(feed intake $\times$ protein content); ERE: $100 \times$ [(dry final body weight $\times$ final energy content) $-$ (dry initial body weight $\times$ initial energy content)]/(feed intake $\times$ energy content). Values are mean $\pm$ SD, within the same column; means with different letters are significantly different ($p < 0.05$).

Within the PMolt crab group, the experimental diets did not significantly influence FCE ($p = 0.73$), PRE ($p = 0.72$) and ERE ($p = 0.19$). In contrast, for the IMolt crab group, FCE, PRE and ERE were significantly higher with diet S100 than with diets F100 and F50S50 ($p < 0.01$).

*3.5. Energy Budget*

The energy budgets were built from the adjusted means calculated for the average feed intakes for each crab group (PMolt and IMolt) fed on the three diet treatments (Table 6). Average feed intakes were 0.65 kJ·dry iBW $g^{-1} \cdot d^{-1}$ and 0.45 kJ·dry iBW $g^{-1} \cdot d^{-1}$ for the PMolt and IMolt crabs, respectively. On average, the PMolt crab group consumed 30% more food than the IMolt group. The higher feed intake in the PMolt crab group led to obviously higher energy loss in feces (FE) and excretion (UE). The lost energy in ecdysis (SE) only concerned the PMolt crab group. The energy allocated for tissue growth (RE) in the PMolt group was 80 to 90% higher than for the IMolt crab group, even though the feed intake was only 30% more for the PMolt group. Finally, the maintenance energy (HEm) was 16 to 21% lower in the PMolt than the IMolt crabs.

**Table 6.** Adjusted mean values of energy budgets, for PMolt and IMolt crabs, fed on 3 different experimental diets.

| Group | Diet | IE | Partition of Energy (kJ·g$^{-1}$ of iBW·d$^{-1}$) | | | | |
|---|---|---|---|---|---|---|---|
| | | | FE | UE | SE | RE | HE$_m$ |
| PMolt | F100 (n = 7) | 0.65 | $0.075 \pm 0.004$ [a] | $0.018 \pm 0.003$ [a] | $0.031 \pm 0.000$ [a] | $0.213 \pm 0.044$ [a] | $0.308 \pm 0.042$ [a] |
| | F50S50 (n = 7) | 0.65 | $0.068 \pm 0.002$ [b] | $0.022 \pm 0.003$ [b] | $0.032 \pm 0.000$ [a] | $0.204 \pm 0.035$ [a] | $0.320 \pm 0.034$ [a] |
| | S100 (n = 7) | 0.65 | $0.067 \pm 0.002$ [b] | $0.016 \pm 0.001$ [a] | $0.032 \pm 0.000$ [a] | $0.246 \pm 0.052$ [a] | $0.282 \pm 0.050$ [a] |
| IMolt | F100 (n = 5) | 0.65 | $0.052 \pm 0.004$ [c] | $0.009 \pm 0.002$ [c] | | $0.024 \pm 0.004$ [b] | $0.367 \pm 0.009$ [b] |
| | F50S50 (n = 5) | 0.65 | $0.043 \pm 0.003$ [d] | $0.013 \pm 0.002$ [d] | | $0.033 \pm 0.014$ [b] | $0.376 \pm 0.011$ [b] |
| | S100 (n = 6) | 0.65 | $0.047 \pm 0.003$ [d] | $0.010 \pm 0.001$ [c] | | $0.056 \pm 0.006$ [c] | $0.345 \pm 0.012$ [c] |

IE: energy intake (kJ·g$^{-1}$ of iBW·d$^{-1}$); FE: feces; UE: excretion; SE: ecdysis; RE: growth; HE$_m$: maintenance. Values are mean $\pm$ SD. Within the same column, different letters indicate significant differences ($p < 0.05$) between the diets within each crab group.

Whatever the crab group considered, the diet treatments had little effect on the energetic balance, except for FE, which was significantly higher for the treatment F100 ($p < 0.01$), and for UE, which was significantly higher for the treatment F50S50 ($p < 0.01$). Furthermore, in the IMolt crab group, the energy allocated to tissue growth was significantly higher for animals fed on the diet S100 ($p < 0.01$). The main differences in the energy budget, calculated as the percentage of energy intake between the IMolt and PMolt crabs, presented in Table 7, were observed for SE (ecdysis), RE (growth) and HEm (maintenance). Only the PMolt crab group was affected by SE, which represents about 5% of the total energy intake. Moreover, RE and HEm were, respectively, 4 times higher and 1.7 times lower in the PMolt than the IMolt crab group.

**Table 7.** Energy budgets in proportion of energy intake (IE): fecal production (FE), excretion (UE), ecdysis (SE), growth (RE) and maintenance ($HE_m$), for PMolt and IMolt crab groups fed on three diet types: F100, F50S50, and S100.

| Group | Diet | %IE | | | | |
|---|---|---|---|---|---|---|
| | | FE | UE | SE | RE | $HE_m$ |
| PMolt | F100 (n = 7) | 11.74 ± 0.52 [a] | 2.83 ± 0.52 [a] | 4.93 ± 0.44 [a] | 33.17 ± 6.98 [a] | 47.90 ± 6.58 [ae] |
| | F50S50 (n = 7) | 10.59 ± 0.38 [b] | 3.39 ± 0.45 [b] | 5.01 ± 0.27 [a] | 31.74 ± 5.65 [a] | 49.84 ± 4.90 [a] |
| | S100 (n = 7) | 10.44 ± 0.48 [b] | 2.48 ± 0.13 [a] | 4.98 ± 0.23 [a] | 38.27 ± 7.57 [b] | 44.16 ± 8.80 [be] |
| IMolt | F100 (n = 5) | 11.53 ± 0.79 [a] | 2.03 ± 0.4 [ce] | | 5.31 ± 0.79 [c] | 81.49 ± 2.02 [cf] |
| | F50S50 (n = 5) | 9.62 ± 0.58 [c] | 2.78 ± 0.35 [d] | | 7.36 ± 3.05 [c] | 83.33 ± 2.45 [c] |
| | S100 (n = 6) | 10.43 ± 0.70 [cb] | 2.33 ± 0.33 [ae] | | 12.33 ± 1.27 [d] | 76.45 ± 2.68 [df] |

Values are mean ± SD. Within the same column, different letters indicate significant differences ($p < 0.05$) between the diets within each crab group.

Diets had a significant effect on FE, EU, RE and HEm. The energy lost in Feces (FE) and excretion (UE) was, respectively, higher for crabs fed the diets F100 and F50S50 ($p < 0.01$). However, those diets marginally affected the global energy budget because FE and EU represent less than 15% of the total energy intake.

The percentage of energy allocated to growth (RE) was 5–7% higher ($p < 0.01$) for crabs fed on S100 than F100 and F50S50, for both PMolt and IMolt crab groups. In contrast, the percentage of energy allocated to the maintenance (HEm) of crabs fed the S100 diet was 4–6% lower than F100 and F50S50 diets, for both PMolt and IMolt crabs.

Finally, the diet treatments had no significant effect on energy lost in exuvia (SE).

## 4. Discussion

Feed for aquaculture requires large quantities of ingredients from the sea and is the primary user of fishmeal among the animal husbandry subsectors [26,27]. Indeed, fishmeal is an important source of quality protein and energy in most aquafeeds and accounted, in 2006, for 68.2% of total global fish meal production [28]. In 1989, fishmeal used in aquaculture accounted for only 10% [29]; this share dramatically increased to 73% by 2010 [26], along with the development of aquaculture.

Historically, the production of carnivorous species, promoted in aquaculture, particularly, marine penaeid shrimps, marine finfish and salmonids, has required the use of fish fishmeal as the main source of protein [30]. The cost of fishmeal is increasing over time as a result of the uncertainty of availability [31]. The authors of [32] indicate that the price of fishmeal increased about 62.9% from August 2005 to June 2008. In such conditions, aquafeed can account for as much as 40–60% of the production cost of the farms [33]. In this framework, it seems useful to assess the mud crab's use of proteins from plants, as a partial or total replacement of fish meal, in formulated feed, in view of its sustainable breeding [34,35].

In our study, the tissue growth of the crabs was considered on a dry weight basis. This approach is quite important as water content of crustaceans is high and varies dramatically

through their molt cycles. Indeed, in crab *S. serrata*, free water content reached 88% in postmolt and decreased to 66% in intermolt and premolt stages [10]. During the molt cycle of the juvenile crab *S. serrata*, two tissue growth periods were identified [9] and confirmed in this study. Thus, in our experimental conditions, two phases of tissue growth were observed: a postmolt period, which lasted between 18 and 22 days, and an intermolt period, which lasted 32 days of the experiment. Most of the tissue growth occurred following crab molting until the early intermolt stage and the remaining during intermolt stage. Indeed, the proportion of organic matter and soluble mineral salt rapidly increased by 80% during the postmolt stage (from ecdysis to C3 stage), and the other 20% was gained during the remaining period (from C4 to D4 stages). This result is consistent with the findings of [10] on the growth study of mud crab juveniles (4.25–7.25 cm carapace width) and confirms our previous results [9]. From our data, and those of [10], we propose schematic growth models of juvenile crab *S. serrata* over two molt cycles where we can distinguish a discontinuous live weight gain at ecdysis, and a continuous tissue growth separated in two successive phases during a molt cycle (Figure 6). In contrast, we noted an extremely rapid increase of mineral salts, corresponding to the mineralization of the cuticle after ecdysis, which is disconnected from tissue growth. Interestingly, we also observed that the feed consumption, or voluntary feed intake (VFI), appears to be correlated with the tissue growth rate. The VFI is maximum during the two weeks following the ecdysis when the tissue growth rate is high and then decreases in parallel to the tissue growth rate until a minimum value shortly before the next molt. In this condition, 50% of the feed is consumed during the first third of the molt cycle when the tissue growth rate is the highest.

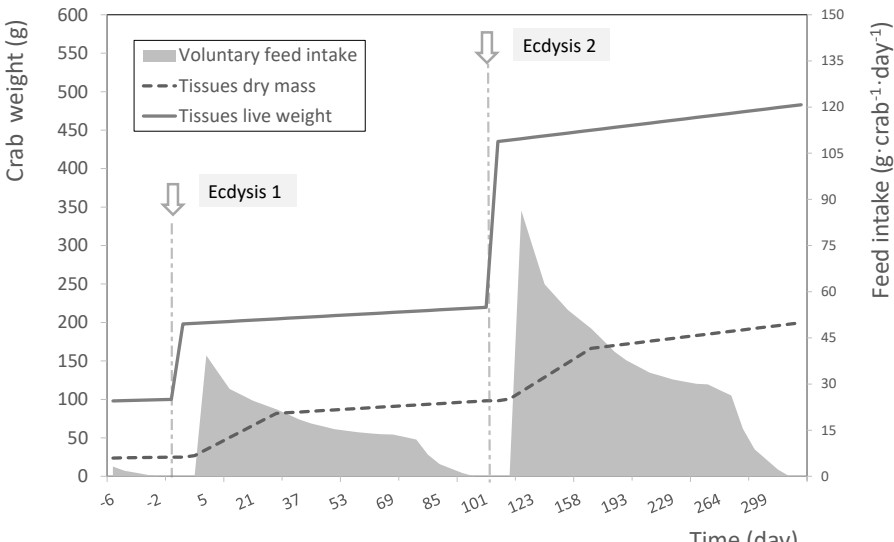

**Figure 6.** Schematic model of growth in live (based on 100 g just before ecdysis 1) and dry mass gain and voluntary food intake (VFI) of juvenile crabs during two molting cycles. The duration of the first molt cycle (MC1) was determined by our study. The duration of the second molt cycle (MC2) was calculated based on the results of [10] and is 1.24 times (+24%) that of MC1. A total of 80% of tissues dry mass growth was observed during the first 30% of the molt cycle, and the remaining 20% during the rest of molt cycle. VFI is dependent on tissues growth, which dramatically increases following the fast tissues growth period and then decrease until the next ecdysis.

In order to compare our results with other studies, we had to transform the data of the latter in order to express them in dry weight. The study of [8] on juvenile crabs *S. serrata* (iBW = 0.25 g) found that the optimum daily protein (0.065 g·dry body g$^{-1}$·d$^{-1}$) and energy (2.351 kJ·dry body g$^{-1}$·d$^{-1}$) intakes were three times higher than that of the present study. These differences could be partly explained by the difference in the size of the crabs used, which was ten times larger in our study. Indeed, it is well admitted that the protein and

energy requirements per weight unit mass of the animal decrease during its growth: for instance, 2 g and 10 g shrimps *L. vannamei* required daily, respectively, 0.014 and 0.007 g·g$^{-1}$ of digested protein and 0.558 and 0.298 kJ·g$^{-1}$·d$^{-1}$ of digestible energy [36].

Many plant-based ingredients are used as a source of protein for aquafeed. Among them, soybean meal (SBM) is well known for its relatively high protein content, well balanced amino acid profile and stable market supply, as well as its reasonable cost [37–39]. The authors of [40] reported that up to 33% fishmeal protein could be replaced by soybean cake for juvenile mitten crabs without reducing growth. The authors of [41] found that 30% of soybean meal and rapeseed meal mixture (1:1 ratio) could replace 40% FM for Chinese mitten crabs without impairing their growth performance and feed utilization. More recently, [42] determined the optimum dietary replacement of fish meal with soy protein concentrate to be 51.49% for the juvenile swimming crab *Portunus trituberculatus*.

However, the inclusion of SBM in fish diets has been limited by relatively high levels of heat stable antinutritional and antigenic factors, including protease inhibitors, oligosaccharides (e.g., stachyose and raffinose), saponins, isoflavones, phytate and tannins [43]. Although heat and enzyme treatments can neutralize some of these compounds, they are still a significant problem when including SBM in aquaculture feeds [44]. While soy protein concentrate (SPC), although more expensive, does not contain the alcohol-soluble fraction present in SBM and has a higher essential amino acid concentration. It also has greater nutrient digestibility compared with SBM and can be included at much higher concentrations in the diets of piscivorous marine species [45,46].

In our study, the juvenile mud crabs digested soybean protein concentrate as well as fishmeal. Moreover, the two ingredients exhibited high digestibility coefficients for dry matter, protein and energy. These results are consistent with previous studies on mud crabs reporting high soybean meal digestibility at an inclusion level of up to 45% with ADC values for ADMD, ACPD and AGED ranging from 80.4–95.7%, 91.7–97.1% and 88.6–97.9%, respectively [5,6,35,47]. Furthermore, [47] showed that crude protein from soybean meal was better digested by mud crabs than the other plant ingredients tested (canola, lupin and cotton seed meal).

Besides the high protein digestibility of SPC for the mud crab, our study showed, under our experimental conditions, that this protein source can also totally replace fishmeal in the diet without affecting the tissue growth of juvenile *S. serrata*. Furthermore, the SPC diet, without fishmeal, led to a better growth rate during intermolt (IMolt crab group) and higher efficiency values of feed conversion, energy retention and protein retention. These results are supported by other authors who worked on crustaceans such as shrimps; the authors of [48] showed that shifting from 80% fishmeal (FM) to soybean and canola meal did not affect the growth rate of *L. vannamei*. Previously, [49] replaced FM with SPC in diets for *L. vannamei* and suggested that SPC, with a methionine supplement, can substitute up to 75% of fishmeal. The authors of [50] showed that the replacement of all the fishmeal by alternative ingredients, such as soy protein concentrate and microbial floc, did not significantly affect the growth, feed conversion ratio, specific growth rate, protein efficiency ratio or the survival of the shrimp *L. vannamei*. In this same shrimp species, [51] have shown that SPC could replace up to 50% of the FM without a significant negative effect on growth performance, hemolymph parameters and immune responses.

Opposite trends have also been observed. The authors of [52] reported that a maximum of 40% of marine protein sources (fishmeal, shrimp head meal and squid meal) could be replaced by soybean meal in diets, but that higher FM replacement levels resulted in lower growth rates of the shrimp *L. vannamei*. Similarly, [53] showed that the maximum FM replacement levels by SPC without affecting the growth rate of the shrimp *Penaeus monodon* were 50%. These authors also reported that high concentrations of soy-based products, in feed, negatively affected the palatability of the diets for shrimps. However, in our study, no influence on the SPC level was noted on the feed intake. Indeed, all experimental diets contained the same amount (20%) of crustacean meal, which is highly palatable for crabs.

The energy budgets of juvenile crabs determined in the present study are based on the average energy intake (IE) for crabs whose tissue growth was studied during the postmolt period (from ecdysis to C2-3 molt stages; PMolt) and the intermolt period (from C3-4 to D1-3 molt stages; IMolt). The adjusted values, for the corresponding average energy intake, were determined for energy lost in feces (FE), excretion (UE), exuveae (SE) and energy retained as growth (RE).

As the experimental unit in this study was the individual crab, it was possible to establish the energy budget for each individual adjusted for the average amount of the energy intake. In the energy budgets of both PMolt and IMolt crab groups, the proportions of the energy intake lost in feces and ammonia excretion were small and similar for the three feed types tested. Generally, the level of nitrogenous excretion is of little importance in the bio-energetic flow for crustaceans [54]. Furthermore, excretion contributed to no more than 3% of the energy budget of fish [55,56], which is similar to our result on crabs. The small proportion of FE and UE in the energy balance of the crabs in our study corresponds to those found by other authors for fishes and shrimps [22,54,57–59].

Based on [20], HEm is defined as the amount of maintenance energy required for an animal to maintain zero energy for growth or an equilibrium between IE and energy expenditure (FE, UE, SE and RE). In the present study, we calculated HEm in two ways: the first by resolving the energy balance equation and the second from the model linking SGR and feed intake when growth was null. After resolving the energy balance equation, HEm values were 0.304 and 0.358 $kJ \cdot dry\ iBWg^{-1} \cdot d^{-1}$ for PMolt and IMolt crabs, respectively. These HEm values included: i) the heat increment of feeding (HiE), ii) the formation and excretion of metabolic wastes, iii) the transformation and inter-conversion of the substrates and their retention in tissues and iv) the molting process [20].

The average HEm value for PMolt and IMolt crabs, estimated from the model linking tissue growth (SGRd) and the feed intake for zero growth (RE = 0), was $0.179 \pm 0.003\ kJ \cdot dry\ iBWg^{-1} \cdot d^{-1}$, which is about half the value of the HEm estimated above by the energy balance equation. This difference is explained by the size of the meals, which was smaller ($0.012\ g.\ dry\ iBWg^{-1} \cdot d^{-1}$) in the case of estimation by the model linking SGR, and the feed intake compared to that obtained by the estimate from the energy balance ($0.03\ g$ and $0.042\ g\ dry\ iBWg^{-1} \cdot d^{-1}$ for PMolt and IMolt crabs, respectively); indeed, the heat increment of feeding increases with the size of the meal [60].

The maintenance energy value of the shrimp *Litopenaeus vannamei* ($345\ kJ \cdot wet\ BWkg^{-1} \cdot d^{-1}$) determined from the growth–meal size model [36] was about six times higher than the HEm value for crabs in our study. This difference could be explained by the fact that the mud crab is much less active than the shrimp, particularly in captivity.

Mud crab juveniles consumed a larger part of the energy for maintenance than for growth, regardless of growth period during the molt cycle and experimental diets. This result is consistent with other studies on fish: the Nile tilapia [57], cobia [22,58] and yellow grouper [59]. However, other studies on crayfish showed a higher proportion of intake energy transferred to growth than to maintenance. Thus, *Cherax tenuimanus* and *C. destructor* could convert, respectively, 41.8–57.5% and 50% of ingested energy into growth [54,61].

One striking result of our study on juvenile mud crabs is the estimated portion of energy intake channeled to molt. Indeed, the energy for molting included energy lost in the exoskeleton (SE) plus the energy required for molting, which in our study was a portion of the maintenance energy (HEm). Our study showed that crabs during intermolt exhibit a higher maintenance energy ratio (up to $83.33 \pm 2.45\%$) than crabs during the postmolt period (up to $49.84 \pm 4.9\%$). The difference between the maintenance energy ratios of the crabs from the two growth periods showed that 33% of energy intake during the intermolt period is stored and required for molting. This proportion included 5% lost with the exoskeleton and the remaining 28% for extra metabolism cost during ecdysis. This result was confirmed by a biochemical analysis indicating significantly higher lipid content in crabs during the intermolt period. Lipid accumulation is completed to anticipate ecdysis, which requires energy, notably, for osmoregulation during molting [62,63]. Similarly, another

study on crabs (*Eriocheir sinensis*) showed that lipids were accumulated in a digestive gland before molting, reaching a peak at late premolt and then decreasing during postmolt [63]. A study on the shrimp *Penaeus semisulcatus* showed the same phenomenon, with lipid drops accumulating in hepatopancreatic R-cells from the intermolt to the premolt stage. Conversely, only a few lipid drops were observed during the postmolt stage [64].

The proportion of energy intake channeled to tissue growth (RE) was lower during the intermolt ($12.32 \pm 0.31\%$) than the postmolt period ($37.39 \pm 6.73\%$). Thus, two interrelated but antagonistic physiological phenomena occurred: ecdysis is necessary for crab growth but it occurs at its expense.

## 5. Conclusions

- Our study on the juvenile crab *S. serrata* led to the following conclusions:
- Voluntary feed intake varies with the rate of the tissue growth rate, which is for a maximum two weeks after ecdysis.
- A total of 33% of the energy intake during the intermolt period are stored for molting.
- When compared with fishmeal, soy protein concentrate did not affect the global energy balance of the crab juveniles.
- Soy protein concentrate can be used as a main source of dietary protein for crab juveniles in captivity.
- Soy protein concentrate sustained good tissue growth and good feed utilization by the crab juveniles, in some cases better than fish meal.

Our results provide useful information on the feed utilization of the crab during a molt cycle. We may also suggest the benefit of studying the nutritional requirements of mud crabs on a dry weight basis, which we believe is more accurate regarding the huge free water change during a molt cycle of this species. Finally, the methodology proposed allows future studies to carry out some specific nutritional experiments in a shorter period without the necessity to wait for several molt cycles.

**Author Contributions:** Conceptualization, all authors N.T.B.N., L.W. and L.C.; methodology, all authors; writing—original draft preparation, N.T.B.N. and L.C.; writing—review and editing, all authors; supervision, L.C. and L.W; project administration, L.W. and L.C.; funding acquisition, L.W. and L.C. All authors have read and agreed to the published version of the manuscript.

**Funding:** This research was funded by the Southern Province of New Caledonia and the MOM project from the French overseas ministry.

**Institutional Review Board Statement:** Ethical review and approval were waived for this study performed on a marine invertebrate other than cephalopods (shrimp) which do not require Ethical review and approval according to EU directive 2010/63/EU (22 September 2010).

**Data Availability Statement:** Not applicable.

**Acknowledgments:** We would like to thank all the IFREMER staff at the aquaculture unit in New Caledonia who kindly helped us with the construction and guiding of the nitrogenous analyses using facilities in the laboratory. We also express our thanks to the staff in the Aquaculture Department of South Province in New Caledonia and the Technical Center for Aquaculture of ADECAL for their help during the running of the experiment.

**Conflicts of Interest:** The authors declare no conflict of interest.

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
