# Peer review of "Feed Efficiency, Tissue Growth and Energy Budget Changes during the Molting Cycle of Juvenile Mud Crab, Scylla serrata: Effects of Dietary Proteins, Fishmeal versus Soy Protein Concentrate"

_fishes, doi:10.3390/fishes7060334_

Round 1

Reviewer 1 Report

Overall a well developed manuscript. The digestibility and N excretion data are all reasonable physiology measures. However, a two-week growth trial from one molt stage to the next is not a nutrition trial. I understand the carbs are held to create soft shells and that is the target of this research, but it is really not a conclusion on nutrition. Lots of good data but downplay the nutritional conclusions and concentrate on the physiology and digestibility. Very good manuscript but could use tempering of some of the statement.

Discussion – comments I am tired of people distorting the facts about fishmeal. Fishmeal is a nutrient source that is in high demand and will always be used by some sector. The aquaculture industry is currently the primary user but it only accounts for around 8% of the diet as an average across all species. It is not the primary protein source and has not been for years for the vast majority of speices. Some sectors use fishmeal as a primary source as nutrition information is lacking for the species. Further, more there is no requirement for ingredients so lets re-write the first paragraph.

Second paragraph – again lets get rid of the wives tails. Carnivorous fish do not require fishmeal. There is no requirement for ingredients, fishmeal is commonly used in high dollar feed and feeds with a high nutrient density. Hence, the perception it that as these feeds require high protein ingredient which is typically met through fishmeal that the fish in turn requires it. There are numerous carnivorous fish that we have taken fishmeal out of the diets. Are you indicating that the crab is carnivorous?

L123 please replace starved with fasted (anywhere it appears)

Figure 6 – where is this from the growth trial was over 32 days this has X axis of 300 day. The molt cycle is not this long. Need to work on this figure and specify source of data.

L450 ish…Please expand and report the level of fishmeal in these diets as well as % replacement. E.g. in a diet containing 40% fishmeal the authors were able to replace 40% reducing the levels to 20% of the diet….or some such. % replacement has little meaning if you do not know how much was in the diet.

L480 I would temper yours statements, the results are a good indication…you did not have enough tissue replacement in this study to make any clear or absolute conclusions on nutrition. Once molt does not make for a successful diet.

L494 ish…If you are going to lean on shrimp there are numerous papers on alternative ingredients. Give me a break you are citing a paper that was published in 1990?

This is a nice paper, good work.

Author Response

Answer to Reviewer 1

  • As rightly suggested by reviewer 1 we have revised the first and second paragraphs of the discussion
  • Fig 6 Indeed we have checked and the figure presented is wrong. We have corrected it and provided additional information on the data used.
  • L450 We agree with the reviewer that the % replacement has little meaning if we don't know how much fish meal was in the feed. However, this information is in the literature cited and is available to the reader. Therefore, we have chosen to simplify our text.
  • L480 As requested by the reviewer, we have put the results into perspective by pointing out that these are observations made under our experimental conditions
  • L494 To our knowledge, the work of Lim and Domini published in 1990 has not been called into question and the results presented are quite relevant, particularly in relation to our own results. In our opinion, the citation of a quality work is scientifically justified even if the publication is old.

Reviewer 2 Report

"The tissue growth can be measured as increase in dry weight, tissue lose water in direct proportion to their gain in dry mass [9, 12]."  This sentence does not make sense - dry mass of tissues can increase regardless of the moisture content.

The experimental rationale is not clear in the introduction as to how the two protein sources are being used experimentally to better understand protein utilisation - is it a graded substitution experiment between the two protein sources?

"intertidal zones associated to with mangroves surrounding the ..."

What is the size range of the juvenile crabs captured and used in the experiments please?

What sort of fishmeal? Source please for Table 1. Same for the soy concentrate, fish oil also please - it is unacceptable to write a feed development paper and not describe your principle ingredients fully.

Why were the crabs only fed for 3 hours every 24 hours?  Most crustaceans, including crabs feed continuously over longer periods such that short feed exposure may skew results by intake and differences in the consistency or cohesiveness of the diets.

The digestibility experiment it is not clear whether this was done on different crabs to the main experimental set? Are there issues with using chromic oxide dietary markers in crustaceans that should be acknowledged?

Why was only ammonia excretion measured? Crustraceans typically have a range of nitrogenous compounds they excrete. 

Can you please analyse the overall differences in feed intake among the three diets and present the results as it appears they intake of the diets was different which may play a key role in some of the other presented feed performance results, which may be due to gustatory stimulation and not differences in dietary nutrient or digestibility attributes.

Personally, I find the introduction to the discussion somewhat dated - there has been a massive sift to the use of plant proteins in aquafeeds over the last decade or more so the aim of this research is in line with this ongoing research and development globally across the sector.

Marked differences in dietary intake with other studies may also be due to differences in attraction of the diets. Crustaceans generally, but especially crabs, are highly chemosensitive and this plays a major role in feed intake, gustatory stimulation and subsequent digestion. The experimental feed formulations included no feed attractants whatsoever, which is unusual for crustacean diets.

Please provide an ethical statement in relation to the experimental animals, especially the euthanasia.

Author Response

Answer to Reviewer 2

  • The sentence « The tissue growth can be measured as….. » has been replaced with « Tissue growth can be measured as the increase in dry mass of the animal, this measure is more relevant than live mass which is subject to the extremely variable water content with the moulting cycle in crustaceans [9, 12]. »
  • « Is it a graded substitution experiment between the two protein sources? This is not a gradual substitution. We compared 3 identical feeds except for the incorporation of fish meal and soluble soy concentrate as protein sources (and wheat as a filler):
  • Feed F100 with 39.28% of protein provided by fish meal and 0% by soybean
  • Feed F50/S50 with 19.64% and 18.36% protein provided by fishmeal and soy respectively
  • Feed S100 with 36.7% of protein provided by soy and 0% by fish.
  • We have change « to mangroves » by « with mangroves »
  • the size of the crab is mentioned on line 125 : « Size rage of the crabs are The initial average body weight and carapace size of crabs were 22.10 ± 8.46 g and 5.06 ± 0.70 cm respectively »
  • Information on fishmeal and soybean meal is now provided in Table 1 A compléter
  • Peruvian fish meal and fish oil (SGS del Peru S.A.C.).
  • Soy protein concentrate ….
  • From experience, we know that the mud crab eats a full meal every 24 hours. In this respect, the feeding behaviour of the crab is very different from that of the shrimp, which feeds continuously. We limit the meal to 3 hours in order to be able to recover the leftovers and measure the ingested food for crabs fed ad libitum, knowing that the crab, in the presence of the meal, finishes eating within 3 hours.
  • The measurement of digestibility is explained in section 2.3.2. This is a measurement carried out in parallel with the main growth experiment.
  • Chromium oxide is widely used as an inert marker to measure the apparent digestibility of foods in insects, terrestrial and aquatic animals.
  • We measured nitrogen « excretion » in the urine and in the faeces (see digestibility measurement). Furthermore, urinary nitrogen excretion in aquatic crustaceans is mainly ammonia and ammonium ions.
  • « analyse the overall differences in feed intake » : In our study we studied growth as a function of meal size (lines 171-172). This method allows us to compare the growth of the animals as a function of the covariate "meal size consumed". The different meal sizes were imposed, the 3 experimental foods were consumed in the same way. Furthermore, the animals under our experimental conditions very well consumed the feed.